# Collagen 1A1 (COL1A1) Is a Reliable Biomarker and Putative Therapeutic Target for Hepatocellular Carcinogenesis and Metastasis

**DOI:** 10.3390/cancers11060786

**Published:** 2019-06-07

**Authors:** Hon-Ping Ma, Hang-Lung Chang, Oluwaseun Adebayo Bamodu, Vijesh Kumar Yadav, Ting-Yi Huang, Alexander T. H. Wu, Chi-Tai Yeh, Shin-Han Tsai, Wei-Hwa Lee

**Affiliations:** 1Department of Emergency Medicine, Taipei Medical University-Shuang Ho Hospital, New Taipei City 235, Taiwan; acls2000@tmu.edu.tw; 2Graduate institute of injury prevention and control, Taipei Medical University, Taipei 110, Taiwan; 3Department of Emergency Medicine, School of Medicine, Taipei Medical University, Taipei 110, Taiwan; 4Department of General Surgery, En Chu Kong Hospital, New Taipei City 235, Taiwan; changhl0321@gmail.com; 5Department of Hematology and Oncology, Cancer Center, Taipei Medical University-Shuang Ho Hospital, New Taipei City 235, Taiwan; 16625@s.tmu.edu.tw (O.A.B.); stardrboypo5210@gmail.com (T.-Y.H.); ctyeh@s.tmu.edu.tw (C.-T.Y.); 6Department of Medical Research & Education, Taipei Medical University-Shuang Ho Hospital, New Taipei City 235, Taiwan; 7The Program for Translational Medicine, Graduate Institute of Biomedical Informatics, College of Medical Science and Technology, Taipei Medical University, Taipei 110, Taiwan; chaw1211@yahoo.com (V.K.Y.); chaw1211@tmu.edu.tw (A.T.H.W.); 8Graduate Institute of Biomedical Informatics, Taipei Medical University, Taipei 110, Taiwan; 9Department of Pathology, Taipei Medical University-Shuang Ho Hospital, 291 Zhongzheng Road, Zhonghe District, New Taipei City 23561, Taiwan

**Keywords:** hepatocellular carcinoma, COL1A1, metastasis, EMT, stemness

## Abstract

Increasing evidence shows that hepatocellular carcinoma (HCC) is a principal cause of cancer-related mortality globally, especially among Asian and African populations. Collagen type I α1 (COL1A1) is the major component of type I collagen. While aberrant expression of COL1A1 and COL1A2 is implicated in numerous cancers, the differential role of COL1A1 in malignant, premalignant and normal tissues remains unclear, and its clinical significance in HCC has not been elucidated. In this study, using bioinformatics analysis of publicly-available HCC microarray data from Gene Expression Omnibus (GEO) and RNAseq data from The Cancer Genome Atlas (TCGA) database, we determined that COL1A1 is significantly upregulated in HCC tumor tissues in comparison to normal tissues. Our analysis also revealed that COL1A1 confers survival advantage and enhanced oncogenicity on HCC cells. Interestingly, the siRNA-mediated silencing of COL1A1 expression (siCOLIA1) suppressed HCC cells clonogenicity, motility, invasiveness and tumorsphere formation. Concomitantly, siCOL1A1 abrogated Slug-dependent epithelial-to-mesenchymal transition (EMT) and HCC stemness gene-signature, by attenuating expression of stemness markers SOX2, OCT4 and CD133. The present study provides some mechanistic insight into COL1A1 activity in HCC and highlights its putative role as an important diagnostic biomarker and potential therapeutic target in early development and metastasis of HCC.

## 1. Introduction

Globally, hepatocellular cancer (HCC) is the fifth most common cause of cancer-related death. The fact that its incidence matches mortality reflects the poor prognosis of this disease [1]. Each year, HCC is diagnosed in more than half a million people worldwide [2]. Despite advances in diagnostic and therapeutic strategies in the last two decades, the management of patients with HCC remains a therapeutic challenge; this is particularly evident in the abysmal median survival time of ≤1 year for patients with advanced HCC, and a five-year relative survival rate ≤9% [3]. Many strategies are available to extend the survival time of liver cancer patients, such as transplantation, surgical resection, target drugs including sorafenib, lenvatinib or regorafenib and immunotherapy (nivolumab) [4]. Surgical resection remains the treatment of choice for patients with well-preserved liver function, and liver transplantation is viewed as the most effective way to improve survival in patients with HCC [5]. Unfortunately, these treatment options do not preclude poor prognosis, while a high risk of postoperative complications and tumor recurrence remain imminent.

Accrued evidence indicate that HCC is a complex and heterogeneous malignancy, with several risks factors, including chronic inflammation secondary to viral infection, like hepatitis B (HBV) and hepatitis C (HCV) virus, alcohol consumption, non-alcoholic steatohepatitis (NASH), bacteria, type 2 diabetes, smoking or chemical. Age and gender are also touted as risk factors for HCC [6]. Among of these risks, HBV and HCV infection constitute a serious public health challenge with approximately 400 and 170 million people with chronic HBV or HCV infection worldwide, respectively [7]. It is estimated that 75% of all HCC cases are due to chronic infection with HBV or HCV [7]. The Asia Pacific region has the largest share of HBV and HCV in the world, with ~74% of global liver cancer-specific deaths occurring in Asia [7]. Patients with HBV and/or HCV infection have a higher predisposition to cirrhosis [8,9]. It has been observed that the occurrence of cirrhosis enhances the risk of HCC, and the presence of cirrhosis is associated with probable death from liver failure or HCC [7,8,9].

The extracellular matrix (ECM) is a complex non-cellular 3D network composed of collagens, proteoglycans/glycosaminoglycans, elastin, fibronectin, laminins, and several other glycoproteins [10]. The major component of the ECM are collagens [11], and type I collagen are found in most connective tissues and embryonic tissues [12]. Typically, type I collagen consists of two chains of collagen type I alpha 1 (COL1A1) and one chain of collagen type I alpha 2 (COL1A2) [13,14]. There is evidence of the involvement of members of the collagen family in carcinogenesis in several tissue types [15], and aberrant expression of *COL1A1* and *COL1A2* has been implicated in some cancers [16,17,18]. Zang et al. [19] reported that *COL1A1* and *COL1A2* were differentially expressed in gastric cancer and predict poor clinical outcomes in gastric cancer patients. However, the expression of *COL1A1* and *COL1A2* in normal epithelium, premalignant and tumor lesions of the stomach is rarely mentioned, and while type I collagen has been associated with hepatic fibrosis [20], there is a dearth of information on the clinical significance of *COL1A1* in patients with HCC, thus, the present study investigate the role of *COL1A1* in HCC.

To unravel the relationship between *COL1A1* expression and/or activity and HCC, we first studied the gene expression profiles in publicly-available whole-genome expression microarray from the Gene Expression Omnibus (GEO) database, accession number GSE14323 comparing gene expression in normal, pre-malignant (cirrhosis) and tumor (HCC) liver tissues. In addition, RNAseq and clinical data were extracted from The Cancer Genome Atlas (TCGA) database for liver hepatocellular carcinoma (LIHC), and further experimental molecular validation was performed using various biomedical assays. COL1A1 expression profile in tumor and normal tissue samples was evaluated as well as the role and molecular mechanism of altered COL1A1 expression in the metastasis and stemness of HCC.

## 2. Materials and Methods

### 2.1. Ethics Approval and Consent to Participate

This study was conducted in a cohort of patients with HCC at Taipei Medical University Shuang-Ho Hospital (TMU-SHH), Taipei, Taiwan. In our representative TMU-SHH HCC samples (*n* = 72), patients were aged from 36 to 81 with a median age of 68.0; 45 were male (62.5%) while 27 were female (37.5%). Based on clinical data extracted from archived patients’ demographic, clinical and biochemical investigation information, clinical characteristics including HBsAg status, cirrhosis, Tumor, Node, Metastasis (TNM) stage, tumor size, and lymph node involvement are also presented in Table 1. The median follow-up time was 26.1 months and 7 patients died during follow-up. This study was approved by the Institutional Human Research Ethics Review Board (TMU-JIRB No. 201302016) of Taipei Medical University.

### 2.2. Analyses of Cancer Microarray and RNAseq Dataset

COL1A1 gene expression profiling and correlative studies were performed using the Gene Expression Omnibus (GEO) human hepatocellular carcinoma microarray dataset with accession numbers GSE14323, GSE3500, GSE14520 and GSE6764 in the Oncomine platform (https://www.oncomine.org/resource/), and TCGA liver cancer hepatocellular carcinoma (LIHC) cohort (*n* = 373). All clinical data were downloaded from the TCGA portal using the University of California Santa Cruz (UCSC) Xena functional genomics explorer (https://xenabrowser.net/heatmap/#) and survival analysis was carried out.

Microarray and RNAseq dataset analyses were performed as previously described [21]. Briefly, after the pre-processing and microarray data statistical analyses using *R* statistical computing environment in the RStudio software version 1.0.143 (https://www.rstudio.com/), we processed the Affymetrix human gene 1.0 ST (HuGene-1_0-st) LIHC array datasets in computable document format (CDF) (*hugene10st_Hs_ENTREZG*), to extract the most complete gene metadata annotation for the Affymetrix probe identifier (IDs). This was followed by data normalization using the Robust Multi-array Average (RMA) algorithm, with log2-transformation and quartile-normalization of the datasets. Where multiple probes had the same Ensembl gene identifier, median gene expression was used. Empirical Bayes-based linear models for microarray data using the limma r-package (http://bioconductor.org/packages/release/bioc/html/limma.html) were employed for identification of differentially expressed genes (DEGs) and Benjamini–Hochberg procedure was used to adjust the *p*-values and reduce the false discovery rate (FDR).

### 2.3. Cell Lines and Cell Culture

The human HBV+ grade IV/V pleomorphic HCC SNU-387 cells, and HBV+ grade II-IV/V HCC SNU-475 cell lines used in the study were purchased from the American Type Culture Collection (ATCC, Manassas, VA, USA). Cells were cultured in Gibco®RPMI 1640 Medium (Cat. No. 11875085, Thermo Fisher Scientific, Inc. Waltham, MA, USA), supplemented with 10% fetal bovine serum (FBS) and 1% penicillin-streptomycin (Cat. No. 15140122, Invitrogen, Thermo Fisher Scientific, Inc. Waltham, MA, USA) at 37 °C, in a 5% humidified CO_2_ incubator. Cells were subcultured at 80–90% confluency.

### 2.4. siRNA-Mediated Knockdown of COL1A1

The ON-TARGETplus COL1A1 siRNA (Cat. No. L-010502-00-0005) with nucleotide sequence TTG GTG TTG TGC GAT GAC GTG used for COL1A1 knockdown was purchased from Dharmacon (Horizon Discovery Group plc, Level Biotechnology, Inc, New Taipei City, Taiwan). The SNU-387 and SNU-475 cells were transfected with the siRNA oligonucleotides and plasmids strictly following manufacturer’s protocol, using Invitrogen^TM^ Lipofectamine® 2000 (Invitrogen Thermo Fisher Scientific, Inc. Waltham, MA, USA).

### 2.5. Western Blotting Analysis

Cellular protein lysates were isolated using the Protein Extraction Kit (QIAGEN, Germantown, MD, USA), and quantified by the Bradford Protein Assay Kit (Beyotime, Beijing, China). An equal amount of (20 μg) of total protein lysate sample was loaded per lane and subjected to sodium dodecyl sulfate polyacrylamide gel electrophoresis (SDS-PAGE). Separated proteins were transferred onto polyvinylidene fluoride (PVDF) membranes, followed by unwanted signal blocking using skim-milk in Tris-buffered saline (TBS), 1× PBS washing 3 times, and then incubation of the PVDF membranes with primary antibodies against COL1A1 (COL1A1 Antibody #84336, 1:1000), E-cadherin (E-Cadherin (24E10) Rabbit mAb #3195, 1:1000), Slug (Slug (C19G7) Rabbit mAb #9585, 1:1000), Vimentin (ab137321, 1:1000), purchased from Santa Cruz (Santa Cruz Biotechnology, Inc, Santa Cruz, CA, USA), and KLF4 (#4038S, 1:1000), OCT4 (#2890S, 1:1000), CD133 (#86781S, 1:1000), and β-actin (#4970S, 1:1000) from CST (Cell Signaling Technology, Inc, Danvers, MA, USA) at 4 °C overnight. The membranes were then incubated in goat anti-rabbit (1:10000; Jackson ImmunoResearch, West Grove, PA, USA) or anti-mouse (1:10000, BD Biosciences, San Jose, CA, USA) horseradish peroxidase (HRP)-conjugated secondary antibodies, and visualized using the enhanced chemiluminescence reagents (ECL, Amersham Biosciences, GE Healthcare, Chicago, IL, USA). The relative band intensity was analyzed using NIH ImageJ software (https://imagej.nih.gov/ij/) and expressed as the ratio of expressed protein to β-actin which served as the loading control. More detailed information about the whole blot can be found in Appendix A.

### 2.6. Colony Formation Assay

To determine the colony-forming ability, 2.5 × 10^3^ siCOL1A1-transfected or wild-type HCC cells were plated in triplicates in 6-well plates (Corning, Corning, NY, USA) consisting of a base layer of 0.5% agarose gel and an upper layer of 0.35% agarose gel with Dulbecco’s Modified Eagle Medium (DMEM)/F-12 medium, N2 supplement, 20 ng/mL of epidermal growth factor (EGF), and basic fibroblast growth factor (bFGF) and incubated for 7 days. The colonies formed were stained with 0.1% crystal violet in 20% methanol, counted and comparative analysis performed. In this study, we defined a colony as a cluster of ≥50 HCC cells.

### 2.7. Wound Healing Migration Assay

Cells were seeded in 6-well plates (Corning, Corning, NY, USA) with Roswell Park Memorial Institute (RPMI) 1640 medium containing 10% FBS and cultured to 95–100% confluence. A scratch along the median axis of the confluent adherent cells layer was then made with a sterile yellow pipette tip. Cell migration, viz-a-viz scratch wound healing images were captured at 0 and 48 h after the scratch was made, under a microscope and analyzed with NIH ImageJ software.

### 2.8. Matrigel Invasion Assay

Cells (2 × 10^5^) were seeded in 24-transwell chambers with an 8 μm pore membrane coated with Matrigel in the upper chamber of the transwell system containing serum-free RPMI 1640 medium. The lower chamber of the transwell chamber contained medium with 20% FBS. After incubation at 37 °C for 6 h, non-invaded HCC cells on the upper side of the membrane were carefully removed with a cotton swab, while the invaded cells were stained with crystal violet dye, air-dried and photographed under a microscope. Images were analyzed with NIH ImageJ software.

### 2.9. Immunohistochemistry

Standard immunohistochemistry (IHC) protocol was used for gene expression profiling and analysis non-tumor liver and HCC tissue samples harvested from the Taipei Medical University-Shuang Ho Hospital HCC cohort (TMU-JIRB No. 201302016). Briefly, 5μm-thick sections were first de-waxed using xylene for 5 min twice and re-hydrated with 100% ethanol twice for 5 min, 95% ethanol for 5 min and 80% ethanol for 5 min, followed by blocking of endogenous peroxidase activity using 3% hydrogen peroxide. Antigen retrieval process was carried out in a pressure cooker where the slides were immersed in 10 mmol/L ethylenediaminetetraacetic acid (EDTA) (pH 8.0) for 3 min, followed by blocking with 10% normal serum. The sections were then incubated with anti-COL1A1 antibody (1:200) overnight at 4 °C, followed by goat anti-mouse IgG HRP-conjugated secondary antibody (1:10,000) from the HRP Polymer Kit (#TP-015-HD; Lab Vision, Fremont, CA, USA). Diaminobenzidine (DAB) was used as the chromogenic substrate, and sections were counterstained with Gill’s hematoxylin (Thermo Fisher Scientific, Waltham, MA, USA).

### 2.10. Sphere Formation Assay

Cells (5 × 10^3^ per well) were plated in ultra-low-attachment six-well plates (Corning) containing stem cell medium consisting of serum-free RPMI 1640 medium supplemented with 10 ng/mL human basic fibroblast growth factor (bFGF) (Invitrogen, Grand Island, NY, USA), 1 × B27 supplement and 20 ng/mL epidermal growth factor (EGF; Invitrogen). The medium was changed every 72 h. After 12 days of incubation, formed spheres were counted and photographs were taken.

### 2.11. Statistical Analysis

All assays were performed at least thrice in triplicate. Values are expressed as the mean ± standard deviation (SD). Comparisons between groups were estimated using Student’s *t*-test for cell line experiments or the Mann–Whitney *U*-test for clinical data, Spearman’s rank correlation between variables, and the Kruskal–Wallis test for comparison of three or more groups. The Kaplan–Meier method was used for the survival analysis, and the difference between survival curves was tested by a log-rank test. Univariate and multivariate analyses were based on the Cox proportional hazards regression model. All statistical analyses were performed using IBM SPSS Statistics for Window, version 20 (IBM, Armonk, NY, USA). A *p*-value <0.05 was considered statistically significant.

## 3. Results

### 3.1. COL1A1 Is Highly Expressed in HCC and Confers Significant Survival Disadvantage

To investigate the *COL1A1* expression profile in HCC and non-tumor liver tissue samples, we used the RNA sequencing and microarray gene profiling data (GSE14323/GPL571, GSE3500, GSE14520 and GSE6764) from GEO and TCGA. The TCGA data included 438 HCC samples, while the GSE14323 had 38 HCC, 58 cirrhosis and 19 normal liver tissue samples. Gene expression profile analysis using heatmap generated from the GSE14323 dataset showed 543 upregulated genes in cirrhosis, and 89 downregulated genes in HCC, with 87 of these genes commonly expressed in both cirrhosis and HCC (Figure 1A,B). Protein–protein interaction (PPI) analysis of the top 21 common genes in cirrhosis and HCC was performed using the STRING functional protein association networks (https://string-db.org), which showed that collagen type genes, including COL1A1, interact with molecular effectors of cytokine signaling in the immune system, namely chemokines CXCR4, CXCL6, CXCL9, CXCL10, CCL19, ANXA1, major histocompatibility complex (MHC) class II proteins HLA-DMA, HLA-DQA1, HLA-DPA1, HLA-DRA, HLA-DQB1, and peptide ligand-binding receptors PTPRC, IFIT1, IFI27 and IFI44L with an average local clustering coefficient of 0.724 and PPI enrichment *p*-value <1.0 × 10^−16^ (Figure 1C). Gene expression analysis of paired tumor-non-tumor samples from TCGA-LIHC cohort (*n* = 438) also showed 1.13-(*p* < 0.001), 1.01-(*p* < 0.01), and 1.07-(*p* < 0.001) fold reduction in COL1A1, COL1A2 and COL4A1 mRNA expression in the HCC tumor compared to the non-tumor ‘normal’ liver samples (Figure 1D). Furthermore, using median gene expression for threshold definition in our Kaplan–Meier survival analysis, we demonstrated that patients with high COL1A1 expression exhibited a worse overall survival (OS) rate with a significant survival disadvantage (~18–26%, *p* = 0.049) compared to the low expression group, however, though statistically insignificant, the high COL1A2 and COL4A1 groups with ~12% (*p* = 0.410) and ~4% (*p* = 0.762) exhibited a survival disadvantage, respectively, compared to the low expression groups (Figure 1E). To corroborate these findings, using the TMU-SHH HCC cohort (*n* = 72), we carried out a Cox proportional hazard analysis of COL1A1 expression and known risk factors for survival in patients with HCC, including age, hepatitis B surface antigen (HBsAg), cirrhosis, tumor size, TNM tumor stage, α-fetoprotein (AFP) and lymph node involvement. Univariate analysis showed that age >60, presence of cirrhosis, tumor size ≥20 mm, AFP ≥400 ng/dL, lymph node metastasis, advanced TNM tumor stage and high COL1A1 expression were risk factors for poor prognosis in the patients with HCC; and interestingly, multivariate analysis revealed that akin to age, cirrhosis, tumor size, AFP level and TNM tumor stage, COL1A1 was an independent prognostic factor of OS in patients with HCC (Table 2).

### 3.2. Upregulated COL1A1 Expression at Both mRNA and Protein Levels Strongly Correlates with Disease Progression

Having established an association between COL1A1 expression profile and patients’ survival, to gain some mechanistic insight, we sought to determine the level of COL1A1 functional activity in the context of HCC progression by probing four GEO datasets, namely the GSE14323 (Mas), GSE3500 (Chen), GSE14520 (Roessler) and GSE6764 (Wurmbach) Liver HCC datasets, as well as immunohistochemistry. We observed that though the expression of COL1A1 mRNA was elevated in pre-malignant cirrhotic liver (4.24-fold, *p* = 1.12 × 10^−10^), it was significantly much higher in the HCC samples (6.08-fold, *p* = 4.54 × 10^−13^), compared to normal liver tissue samples from the GSE14323 cohort (Figure 2A). This markedly elevated COL1A1 mRNA expression profile was replicated in GSE3500 (*p* = 0.01), GSE14520 (*p* = 4.26 × 10^−4^) and GSE6764 (*p* = 1.68 × 10^−6^) HCC samples, compared to their normal counterparts (Figure 2B). We also demonstrated that relative to the non-tumor tissues, COL1A1 protein expression was over-expressed in the HCC samples in a stage-dependent manner (Figure 2C). These results do not only indicate enhanced transcriptional and post-translational activity of COL1A1 in patients with HCC, but also reveal a strong correlation between elevated COL1A1 expression and disease progression.

### 3.3. Knockdown of COL1A1 Suppressed HCC Cell Migration and Invasion through Deregulated Epithelial-to-Mesenchymal Transition (EMT), In Vitro

Since enhanced migration and invasion are characteristic of metastatic or late-stage disease [22], we investigated the probable role of COL1A1 in the enhanced metastatic phenotype of HCC cells using siRNA-mediated transient loss of COL1A1 function in human HBV+ grade IV/V pleomorphic HCC SNU-387 cells, and HBV+ grade II–IV/V HCC SNU-475 cells. With knockdown efficacy of 41% and 67% in the SNU-387 and SNU-475 cells (Figure 3A), reduced COL1A1 expression (siCOL1A1) markedly suppressed the number of invaded cells in both SNU-387 (~70%, *p* < 0.05) and SNU-475 (~58%, *p* < 0.05) cells, compared to the control cells (Figure 3B). Similarly, our wound-healing migration assay results showed that siCOL1A1 attenuated the migration capabilities of SNU-387 and SNU-475 by 62% (*p* < 0.05) and 55% (*p* < 0.05), respectively, compared to the control cells, after 48 h (Figure 3C). Understanding the critical role of epithelial–to–mesenchymal transition (EMT) the acquisition of the metastatic phenotype [22], our Western blot analysis of the effect of silencing COL1A1 on EMT markers Slug, vimentin, and E-cadherin demonstrated that siCOL1A1 significantly co-suppressed COL1A1, Slug and vimentin protein levels, while conversely upregulating the expression of E-cadherin protein in both SNU-387 and SNU-475 cell lines (Figure 3D). These results, at least in part, are indicative of a critical role for COL1A1 in the induction and/or enhancement of HCC cells invasion and migration in vitro through EMT deregulation.

### 3.4. COL1A1 Is Strongly Associated with and Is a Probable Bridge between the Metastatic and Cancer Stem Cell-Like Phenotypes of HCC

Having established a critical role for COL1A1 in the induction and/or enhancement of HCC cells invasion and migration through EMT deregulation, in vitro, we investigated probable links between COL1A1-induced metastatic phenotype and CSC-like phenotype using mRNA data (RNA Seq V2) from the provisional TCGA LIHC cohort consisting of 371 patients and 373 samples. Results of our Spearman correlation analysis revealed strong correlation between COL1A1 and metastasis markers, vimentin (VIM) (0.79 (*p* = 1.25 × 10^−79^)), snail (SNAI1) (0.75 (*p* = 6.21 × 10^−68^)), matrix metalloproteinase (MMP)2 (0.87 (*p* = 1.18 × 10^−116^)), MMP9 (0.40 (*p* = 8.29 × 10^−16^)), TWIST1 (0.67 (*p* = 6.53 × 10^−50^)) (Figure 4A), as well as markers of cancer stemness, CD144/PROM1 (0.47 (*p* = 1.56 × 10^−21^)), CD44 (0.39 (*p* = 3.03 × 10^−15^)), KLF4 (0.37 (*p* = 1.46 × 10^−13^)) and ABCC1 (0.50 (*p* = 1.39 × 10^−24^)) (Figure 4B). However, we found no significant correlation between COL1A1 and SOX2 (0.07 (*p* = 0.193)) (Figure 4B). These bioinformatics-based statistical data do indicate a strong association between COL1A1, markers of metastasis and cancer stemness, as well as suggest a role for COL1A1 in the modulation of HCC CSCs-like phenotype.

### 3.5. Loss of COL1A1 Function Significantly Impair Colony and Tumorsphere Formation of HCC Cells In Vitro

Based on our earlier results and the understanding that enhanced invasiveness and clonogenicity characterize CSCs, with circulating liver CSCs playing active roles in the preparation of new sites for colonization [23,24], we evaluated the effect of siCOL1A1 on the clonogenic and stemness phenotypes of HCC cells using SNU-387 and SNU-475. Results from our colony formation assays demonstrate that the number of colonies formed was significantly reduced in the siCOL1A1-transfected SNU-387 (3.2-fold, *p* < 0.05) and SNU-475 (2.4-fold, *p* < 0.05) cells in comparison to the control wild-type cells (Figure 5A). In addition, we demonstrated that silencing COL1A1 caused profound loss of ability to form in vitro liver CSC models, primary hepatospheres in both the siCOL1A1-transfected SNU-387 (2.9-fold, *p* < 0.01) and SNU-475 (2.4-fold, *p* < 0.01) cells in comparison to the control wild-type cells. Interestingly, aside from the decreased tumorsphere formation efficacy, quantitatively and qualitatively, siCOL1A1 suppressed the self-renewal capability of the HCC cells, as demonstrated by an 89% (*p* < 0.01) and 71.4% (*p* < 0.01) reduction in the ability of disintegrated primary SNU-387 or SNU-475 tumorspheres to form secondary hepatospheres (Figure 5B). Expectedly, this siCOL1A1-induced suppressed tumorsphere formation efficacy and loss of self-renewal phenotype was associated with concurrent downregulation of COL1A1, KLF4, OCT4, YAP1 and CD133 in the siCOL1A1 HCC cell lines, compared to their wild-type counterparts (Figure 5C). Furthermore, for translational relevance, using the functional characterization and druggability studies of COL1A1, associated co-expressed MHC class I and II proteins, cell survival genes and HCC stemness markers, we demonstrated that COL1A1is a highly druggable oncogene that is overexpressed in patients with HCC, as found in the SAMSUNG HCC (*n* = 6,619), Asian Cancer Research Group (ACRG) HCC (*n* = 88) and TGCA LIHC (*n* = 373) cohorts (Figure 5D).

## 4. Discussion

The development of synchronous or metachronous metastasis by patients with HCC even post curative surgical resection is a medical challenge in liver oncology clinics. While the majority of these cases are intrahepatic metastatic, it is estimated that about 13.5–42% are extrahepatic metastasis for patients with HCC [25,26,27,28]. The relatively limited therapeutic options for managing metastatic and late stage HCC [25,26,27,28] necessitates the discovery of reliable targetable or druggable disease-specific biomarkers such as COL1A1, and gives clinical and/or translational relevance to the findings of this present study. The present study provides preclinical evidence indicating that COL1A1 is a reliable biomarker and putative therapeutic target for hepatocellular carcinogenesis and metastasis by demonstrating that (i) COL1A1 is highly expressed in HCC and confers significant survival disadvantage, and that (ii) upregulated COL1A1 expression at both mRNA and protein levels strongly correlates with disease progression. We also showed that the (iii) knockdown of COL1A1 suppressed HCC cell migration and invasion through deregulated EMT, in vitro. Finally, we demonstrated that (iv) COL1A1 is strongly associated with and is a probable bridge between the metastatic and CSCs-like phenotypes of HCC, and that the (v) loss of COL1A1 function significantly impair colony and tumorsphere formation of HCC cells in vitro.

Our findings that COL1A1 is highly expressed at both mRNA and protein levels in HCC, strongly correlates with disease progression, and confers significant survival disadvantage to patients with HCC ( Figure 1 and Figure 2). It is consistent with recent findings showing that COL1A1, activated by multiple signaling pathways in a myocardin-related transcription factor A (MRTFA)-mediated manner, is highly expressed in human breast cancer [29], additionally COL1A1 and COL1A2 mRNA levels are significantly upregulated in gastric cancer tissues compared to normal tissues, and lower COL1A1 and COL1A2 expression levels are strongly associated with better overall survival in patients with gastric cancer [30]. However, this demonstrated implication of high COL1A1 expression in poor prognosis among patients with HCC contradicts findings from an isolated study suggesting that COL1A1 gene expression is significantly decreased in HCC samples and that this tumor associated suppressed COL1A1 mRNA expression levels significantly correlated with poor OS [18]. While we cannot fully rationalize this contrasting conclusion, it is worth pointing out that in the same study, COL1A1 methylation and expression analysis of the “study patient” and HCC cell lines showed apparent enhancement of COL1A1 methylation status, mRNA and protein expression levels; this is antithetic to the study conclusions and relevant in the context of contemporary knowledge wherein DNA methylation not only predisposes tissue cells to tumorigenesis, but is increasingly shown to contribute to the tumor state via repression of tumor suppressor genes and inhibition of cellular differentiation plasticity [31,32]. By same token, there is evidence that persistent methylated DNA after tumor resection indicates residual disease and is associated with disease recurrence and poor prognosis (reviewed in [32]).

Furthermore, concordant with the accruing evidence that enhanced expression of COL1A1 promotes metastasis of breast cancer [33], augments the proliferation and invasion of gastric cancer cells by downregulating miR-129-5p [34], and via dysregulation of the wingless-related integration site/planar cell polarity (WNT/PCP) pathway, promotes metastasis in colorectal cancer (CRC), our finding that the knockdown of COL1A1 suppressed HCC cell migration and invasion by downregulating Slug and vimentin, while upregulating E-cadherin, in vitro (Figure 3) is not out of place. In fact, additional corroboration of our findings is found in the work of Toiyama et al. [35] showing that while Slug and vimentin are highly expressed in higher tumor (T) stage, lymph-node involvement, hepatic metastasis and advanced TMN stage, siRNA-silencing of Slug in CRC cells induced reduction in cell proliferation, suppressed EMT, and attenuated the invasion and migration in CRC cells.

In line with recent findings that enhanced invasiveness and clonogenicity characterize CSCs, and that circulating liver CSCs play critical roles in the preparation of new sites for malignant colonization [23,24], and having demonstrated that COL1A1 plays a critical role in the induction and/or enhancement of HCC cells invasion and migration through the deregulation of EMT, in vitro, we investigated probable links between COL1A1-induced metastatic phenotype and CSCs-like phenotype. Our results revealed that COL1A1 is strongly associated with and is a probable bridge between the metastatic and CSCs-like phenotypes of HCC (Figure 4), which is particularly interesting and bears translational relevance against the background that metastatic disease is implicated in over 90% of cancer related mortalities, and in the light of the role of CSCs in the self-renewal of cancerous cells, resistance to anti-cancer therapy, metastatic dissemination to secondary sites and disease recurrence [36]. Based on our findings, we posit that aberrant expression of COL1A1 confer enhanced clonal survivability on enriched cancerous cells and drive the acquisition of cellular traits that favor the formation and dissemination of micro- and macro-metastases culminating the intra-tumoral Darwinian evolutionary bioprocess. This position is reinforced by the fact that the loss of COL1A1 function significantly impairs colony and tumorsphere formation of HCC cells in vitro (Figure 5), and consistent with current therapeutic strategies that target CSCs-like HCC cells with enhanced capacity to colonize distant organs and exert tumor-initiating potential in the tumor microenvironment [22]. Interestingly, using bioinformatics-based algorithms for functional characterization and druggability studies, we also showed that COL1A1 is a targetable or druggable oncogene that not only has a high degree of gene neighborliness, but is a probable modulator of other collagen types such as COL1A2, COL4A1, and COL4A5, survival genes ABL1, ABL2, and PPARG, as well as known markers of cancer stemness, namely ALDH1, CD44, CD133/PROM1, KLF4, MYC, OCT4/POU5F1 and SOX2, in patients with HCC (Figure 5).

## 5. Conclusions

In conclusion, our pictorial abstract (Figure 6) presents preclinical evidence that aberrant expression of COL1A1 is a putative biomarker of HCC initiation and progression and targeting COL1A1 elicits abrogation of HCC resistance to anticancer therapy, metastatic dissemination to secondary sites, self-renewal and disease recurrence.

## Figures and Tables

**Figure 1 cancers-11-00786-f001:**
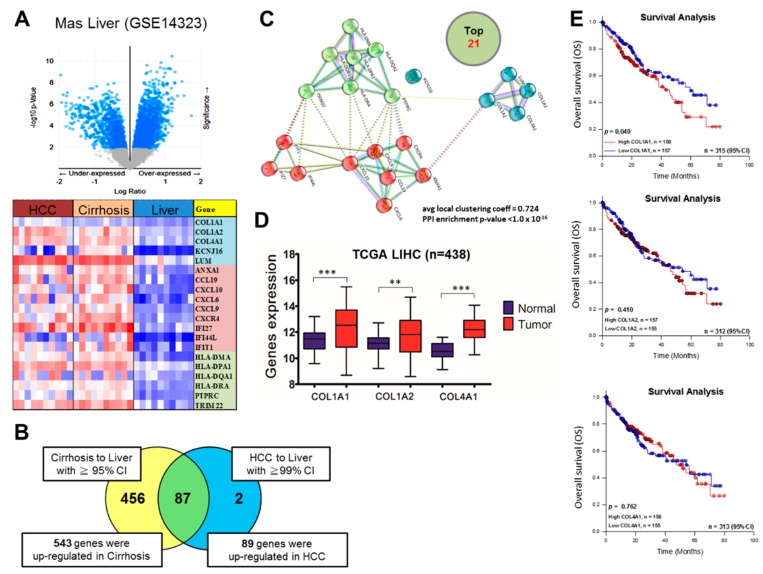
COL1A1 is highly expressed in hepatocellular carcinoma (HCC) and confers a significant survival disadvantage. (**A**) Volcano plot (upper panel) and heat map (lower panel) showing the differential expression of different genes expression in HCC, cirrhosis and normal liver tissue samples from the GSE14323 dataset. (**B**) Venn diagram showing that of the 543 and 89 genes upregulated in cirrhosis and HCC, respectively, 87 genes were common to both cirrhosis and HCC. (**C**) Functional protein–protein interaction of the top 21 of 87 co-upregulated genes as depicted by Search Tool for the Retrieval of Interacting proteins database (STRINGdb). (**D**) Graphical representation of the differential expression of COL1A1, COL1A2 and COL4A1 in normal liver or HCC samples from The Cancer Genome Atlas (TCGA) liver cancer hepatocellular carcinoma (LIHC) (*n* = 373). (**E**) Kaplan–Meier plots of the effects of high or low expression of COL1A1 (upper), COL1A2 (middle) or COL4A1 (lower) on the survival rates in the TCGA LIHC cohort. Median gene expression was used to determine low/high cut-off value. ** *p* < 0.01, *** *p* < 0.001.

**Figure 2 cancers-11-00786-f002:**
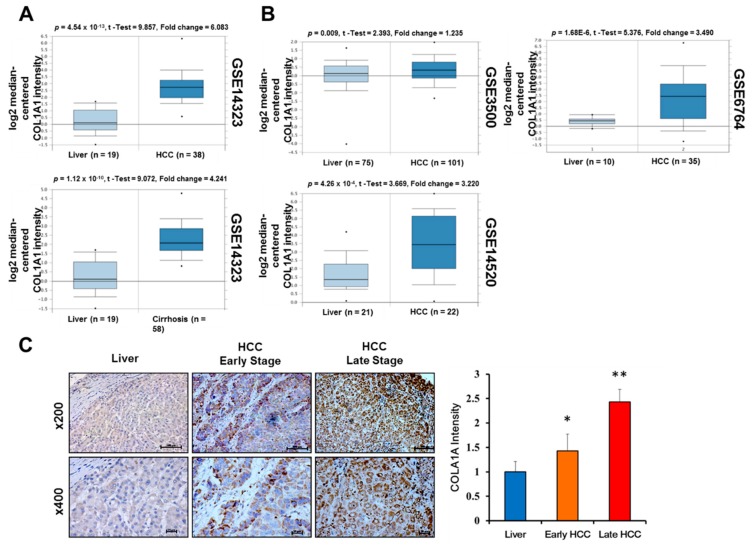
Upregulated COL1A1 expression at both mRNA and protein levels strongly correlates with disease progression. (**A**) Differential expression of COL1A1 in HCC (upper panel) and cirrhosis (lower panel) compared to normal liver samples in the GSE14323 Mas liver dataset. (**B**) COL1A1 is highly expressed in GSE3500 Chen liver, GSE14520 Roessler liver and GSE6764 Wurmbach liver HCC specimens compared to non-tumor liver specimens. (**C**) Representative IHC photo-image (left) and graphical quantification (right) of COL1A1 protein expression in early stage, late stage HCC tissues and adjacent normal tissues. * *p* < 0.05, ** *p* < 0.01.

**Figure 3 cancers-11-00786-f003:**
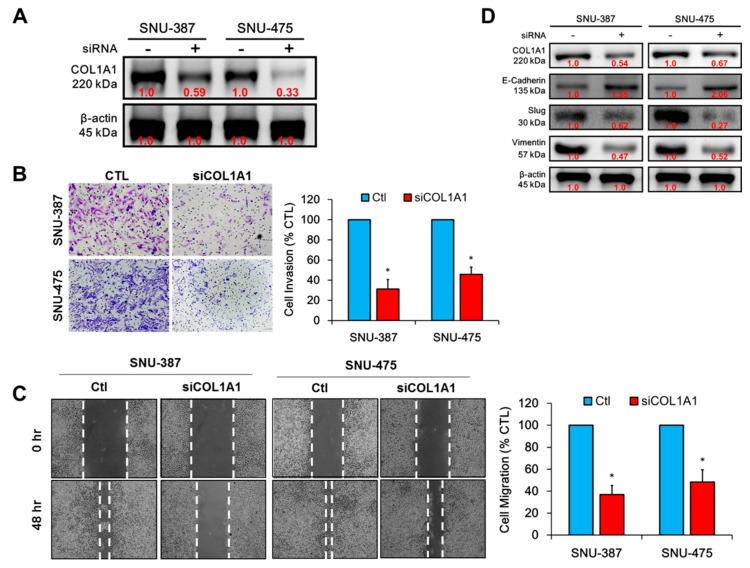
Knockdown of COL1A1 suppressed HCC cell migration and invasion through deregulated epithelial–to–mesenchymal transition (EMT), in vitro. (**A**) Western blot images showing the knockdown efficacy of COL1A1 in SNU-387 or SNU-475 cells. Representative photo-images (left) and histograms (right) of the effect of siCOL1A1 on the (**B**) invasion (scale bar = 100 µm, original magnification ×200) and (**C**) migration of SNU-387 and SNU-475 cells, magnification ×200. (**D**) Representative Western blot images of the effect of siCOL1A1 on the expression levels of COL1A1, E-cadherin, Slug and vimentin protein in SNU-387 and SNU-475. β-actin was used as a loading control. * *p* < 0.05; Ctl, wild type cells not transfected with siCOL1A1.

**Figure 4 cancers-11-00786-f004:**
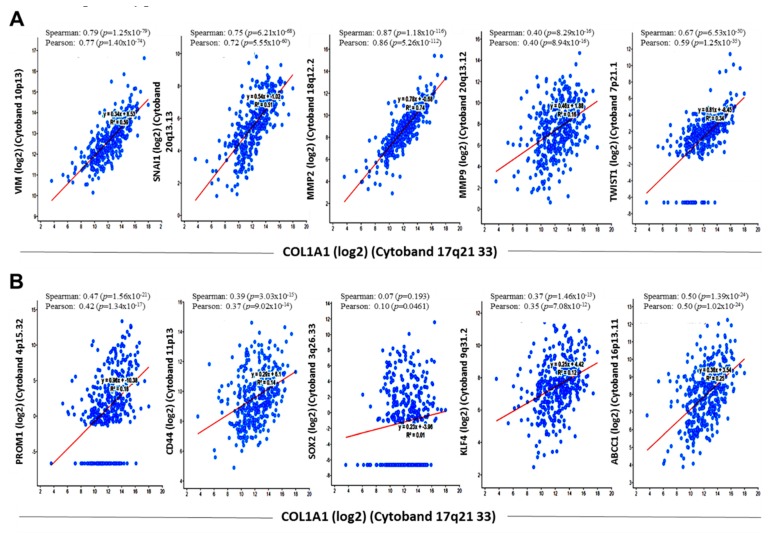
COL1A1 is strongly associated with and is a probable bridge between the metastatic and CSC-like phenotypes of HCC. Scatter plot showing the correlation between the expression of COL1A1 mRNA and that of (**A**) VIM, SNAI1, MMP2, MMP9 and TWIST, as well as (**B**) PROM1, CD44, SOX2, KLF4 and ABCC1 mRNAs in the provisional TCGA LIHC mRNA data (RNA Seq V2) cohort (371 patients/373 samples).

**Figure 5 cancers-11-00786-f005:**
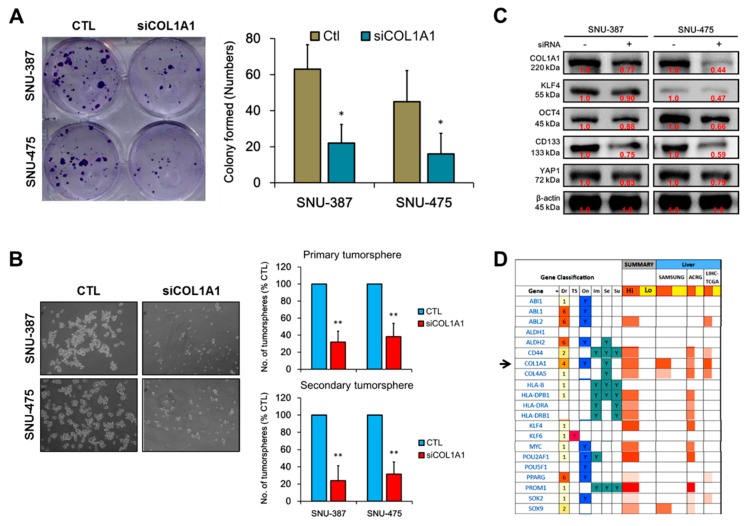
Loss of COL1A1 function significantly impairs colony and tumorsphere formation of HCC cells in vitro. Representative photo image (left) and histograms (right) showing the inhibitory effect of siCOL1A1 on the (**A**) colony formation potential and (**B**) number and size of primary and secondary hepatospheres formed by SNU-387 or SNU-475 cells, scale bar = 50 μm, original magnification ×200. (**C**) Western blot images of the effect of siCOL1A1 on the expression level of COL1A1, KLF4, YAP1, OCT4 and CD133 proteins. (**D**) Functional characterization and druggability studies using the SAMSUNG HCC, Asian Cancer Research Group (ACRG) HCC and TCGA LIHC cohorts. Arrowhead, gene of interest; Dr, small molecule drugability score; TS, tumor suppressor; On, oncogene; Im, involved in immune modulation; Se, secreted gene; Su, cell surface protein. β-actin was used as a loading control. * *p* < 0.05, ** *p* < 0.01; Ctl, wild type cells not transfected with siCOL1A1.

**Figure 6 cancers-11-00786-f006:**
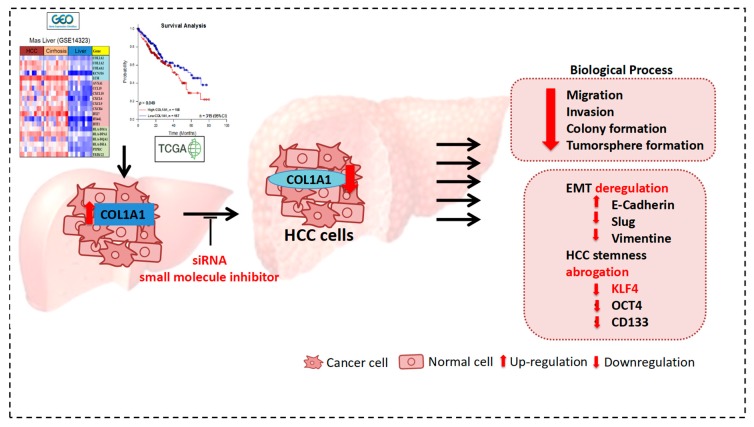
Pictorial abstract showing that the therapeutic or molecular targeting of COL1A1 function significantly impairs the oncogenicity and cancer stem cell-like phenotype of HCC cells by deregulating EMT and abrogating HCC stemness.

**Table 1 cancers-11-00786-t001:** Patient clinicopathological characteristics of TMU-SHH HCC cohort. * *p* < 0.05.

Clinicopathological Variable	No.	High COL1A1	Low COL1A1	*X* ^2^	*p*-Value
**Age, years**					
≤60	33	19	14	0.014	0.905
>60	39	23	16		
**Gender**					
Male	45	21	24	0.533	0.465
Female	27	15	12		
**HBsAg**					
negative (−)	11	4	7	2.578	0.108
positive (+)	61	38	23		
**α-fetoprotein** (AFP)					
<400 ng/dL	25	12	8	6.254	0.017 *
≥400 ng/dL	47	30	22		
**Cirrhosis**					
Absent	23	9	14	4.374	0.036 *
Present	49	32	17		
**Tumor size (mm)**					
<20	19	8	11	1.891	0.169
≥20	53	32	21		
**Lymph node metastasis**					
negative (−)	31	13	18	5.002	0.025 *
positive (+)	41	28	13		
**TNM tumor stage**					
I+II	29	10	19	4.677	0.031 *
III+IV	43	26	17		

**Table 2 cancers-11-00786-t002:** Univariate and multivariate analysis of COL1A1 expression in TMU-SHH HCC cohort.

Clinicopathological Variables	Univariate Analysis			Multivariate Analysis		
	HR	95% CI	*p*-Value	HR	95% CI	*p*-Value
**Age** (≤60 vs >60)	0.401	0.215–1.003	0.037 *	0.649	0.776–1.019	0.041 *
**Gender** (male vs female)	1.049	0.630–1.315	0.764			
**HBsAg** (positive vs negative)	0.414	0.206–0.983	0.071	0.705	0.374–1.028	0.673
**Cirrhosis ^#^**	1.013	0.881–1.045	<0.001 *	1.009	0.796–1.042	<0.001 *
**Tumor size (mm) ^#^**	1.430	1.030–1.719	<0.001 *	1.602	1.343–2.152	0.021 *
**α-fetoprotein (AFP) ^#^**	2.316	1.683–3.473	<0.001 *	2.016	1.476–2.861	<0.001 *
**Lymph node metastasis ^#^**	0.875	0.652–0.967	0.017 *	0.784	0.569–1.029	0.849
**TNM tumor stage ^#^**	1.543	1.241–2.068	<0.001 *	1.314	1.050–1.826	0.010 *
**COL1A1 expression** (high vs. low)	2.008	0.644–5.316	<0.001 *	2.252	0.971–6.473	<0.001 *

^#^ Cirrhosis: cirhosis vs. normal or fibrosis; Tumor size (mm): <20 vs. ≥20; Lymph node metastasis: positive vs. negative; AFP: <400 ng/dL vs. ≥ 400 ng/dL; TNM tumor stage: III–IV s. I–II; * Significant by Cox regression using the proportional hazard model.

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
