# Peer review of "Collagen 1A1 (COL1A1) Is a Reliable Biomarker and Putative Therapeutic Target for Hepatocellular Carcinogenesis and Metastasis"

_cancers, 2019, doi:10.3390/cancers11060786_

Round 1

Reviewer 1 Report

I have read with interest this manuscript. The authors found that the overexpression of COL1A1 at both levels (mRNA and protein) was significantly associated with worse prognosis in patients affected by HCC. I find this evidence interesting and quite novel. The manuscript is overall well-written and it has well presented.

I would like to ask to the authors if they can add a multivariate analysis by including all those clinico-pathological factors that traditionally are prognostically important. I refer to: number and size of tumor, micro- and macro-vascular invasion, tumor grading, quality of the underlying liver parenchyma (normal liver vs. chronic hepatitis/cirrhosis), level of AFP, etc... plus the expression of COL1A1.

Author Response

Point-by-point responses to reviewer’s comments:

We would like to thank the reviewers for the thorough reading of our manuscript as well as their valuable comments. We have followed their comments closely and feel that they have further strengthened the manuscript. Below are our point-by-point responses.

Q1: Reviewer #1:   I have read with interest this manuscript. The authors found that the overexpression of COL1A1 at both levels (mRNA and protein) was significantly associated with worse prognosis in patients affected by HCC. I find this evidence interesting and quite novel. The manuscript is overall well-written, and it has well presented.

A1: We sincerely thank the reviewer for the time taken to review our work, and for the suggestions given. In this revised manuscript, we have made use of the reviewer’s suggestion.

Q2: Reviewer #1: In I would like to ask to the authors if they can add a multivariate analysis by including all those clinico-pathological factors that traditionally are prognostically important. I refer to: number and size of tumor, micro- and macro-vascular invasion, tumor grading, quality of the underlying liver parenchyma (normal liver vs. chronic hepatitis/cirrhosis), level of AFP, etc... plus the expression of COL1A1.

A2: We thank the reviewer for this insightful suggestion. As suggested by the reviewer, we have now provided data for Cox univariate and multivariate analysis in our revised manuscript. Please kindly see our newly Tables section, Tables 1 and 2.

Table 1. Patient clinicopathological characteristics of TMU-SHH HCC cohort

Table 2. Univariate and multivariate analysis of COL1A1 expression in TMU-SHH HCC cohort

Please also kindly see our revised Materials and Methods section, Pages 2-3, Lines 92-100.

2.1. Ethics approval and consent to participate

This study was conducted in a cohort of patients with HCC at Taipei Medical University Shuang-Ho Hospital, Taipei, Taiwan. In our representative TMU-SHH HCC samples (n = 72), patients aged from 36 to 81 with median age of 68.0; 45 were male (62.5%) while 27 were female (37.5%). Based on clinical data extracted from archived patients' demographic, clinical and biochemical investigation information, clinical characteristics including HBsAg status, cirrhosis, TNM stage, tumor size and lymph node involvement are also presented in Table 1. The median follow-up time was 26.1 months and 7 patients died during follow-up. This study was approved by the Institutional Human Research Ethics Review Board (TMU-JIRB No. 201302016) of Taipei Medical University.

Also kindly see our revised Results section, Pages 5-6, Lines 204-235.

3.1. COL1A1 is highly expressed in HCC and confers significant survival disadvantage

To investigate the COL1A1 expression profile in HCC and non-tumor liver tissue samples, we used the RNA sequencing and microarray gene profiling data (GSE14323/GPL571, GSE3500, GSE14520 and GSE6764) from GEO and TCGA. The TCGA data included 438 HCC samples, while the GSE14323 had 38 HCC, 58 cirrhosis and 19 normal liver tissue samples. Gene expression profile analysis using heatmap generated from the GSE14323 dataset showed 543 up-regulated genes in cirrhosis, and 89 down-regulated genes in HCC, with 87 of these genes commonly expressed in both cirrhosis and HCC (Figure 1A, B). Protein-protein interaction (PPI) analysis of top 21 common genes in cirrhosis and HCC was performed using the STRING functional protein association networks (https://string-db.org), showed that collagen type genes, including COL1A1 interact with molecular effectors of cytokine signaling in the immune system, namely chemokines CXCR4, CXCL6, CXCL9, CXCL10, CCL19, ANXA1, major histocompatibility complex (MHC) class II proteins HLA-DMA, HLA-DQA1, HLA-DPA1, HLA-DRA, HLA-DQB1, and peptide ligand-binding receptors PTPRC, IFIT1, IFI27 and IFI44L with an average local clustering coefficient of 0.724 and PPI enrichment p-value <1.0e-16 (Figure 1C). Gene expression analysis of paired tumor-non-tumor samples from TCGA-LIHC cohort (n = 438) also showed 1.13- (p<0.001), 1.01- (p<0.01), and 1.07- (p<0.001) fold reduction in COL1A1, COL1A2 and COL4A1 mRNA expression in the HCC tumor compared to the non-tumor ‘normal’ liver samples (Figure 1D). Furthermore, using median gene expression for threshold definition in our Kaplan-Meier survival analysis, we demonstrated that patients with high COL1A1 expression exhibited worse overall survival (OS) rate with a significant survival disadvantage (~18 - 26%, p = 0.049) compared to the low expression group, however, though statistically insignificant, the high COL1A2 and COL4A1 groups with ~12% (p = 0.410) and ~4% (p = 0.762) survival disadvantage, respectively, compared to the low expression groups (Figure 1E). To corroborate these findings, using the TMU-SHH HCC cohort (n = 72), we carried out a Cox proportional hazard analysis of COL1A1 expression and known risk factors for survival in patients with HCC, including age, hepatitis B surface antigen (HBsAg), cirrhosis, tumor size, TNM tumor stage, α-fetoprotein (AFP), and lymph node involvement. Univariate analysis showed that age > 60, presence of cirrhosis, tumor size ≥20 mm, AFP ≥ 400ng/dL, lymph node metastasis, advanced TNM tumor stage, and high COL1A1 expression were risk factors for poor prognosis in the patients with HCC; and interestingly, multivariate analysis revealed that akin to age, cirrhosis, tumor size, AFP level, and TNM tumor stage, COL1A1 is an independent prognostic factor of OS in patients with HCC (Table 2).

Reviewer 2 Report

In the study, the author identified COL1A1 as a survival-related biomarker and therapeutic target in HCC and its metastasis.  Combining the deep data mining from TCGA, clinic samples check and cell in vitro study, the work showed high level of COL1A1 has a correlation to HCC and its metastasis. However, there are some important major and minor issues that should be fixed before this article could consider for publication in Cancers.

Some minor questions,

1.       The author should better add the dilution of primary antibody used in western blot study part.

2.       Some writing might be modified, such as Protein-protein interaction (PPI) interaction analysis, people usually wrote as Protein-protein interaction (PPI) analysis.

3.       Meanwhile, about the PPI network analysis, could the author try 2 to 3 different interactome analysis tool to verify the results or confirm the top genes interaction?

4.       2.2 Analyses of cancer microarray and RNAseq dataset, could the author add a little more detail on the data extraction and analysis?

5.       About 2.6 colony formation assay, what’s the cell concentration plated in the 6 well plates? The author should better add a little more detail here.

6.       In line 279, MMP2 (0.40 (p = 8.29e-16)), should be MMP9 (0.40 (p = 8.29e-16)). Please check line 170, (bFGF; (Invitrogen, Grand Island, NY, USA),

Some major questions detailed as following,

1.       Please add detail on IHC staining, including the primary and secondary antibody dilution, incubation time, et al. meanwhile, could the author put clinic sample data detail as supplementary table in the manuscript, including patient age, sex, risk factors (virus infection, or alcohol or NASH etc)?

2.       About TCGA data, did the author combine all various HCC samples, if choose only one, there usually showed about 373- 377 samples in total?

3.       Please add high resolution picture of the figure 1c. and the figure legend 1 E is not clear, the author should better modify it. About the figure 1B, could the author put the full gene list as supplementary table in the manuscript?

4.       To figure 2C, could the author double check the figures to make sure the figures used correctly, especially on the HCC late stage.

5.       To the figure 3 study, what’s the control, the author should better marked clearly, is it the cell treated with scramble siRNA or the cell without any treatment?  There are similar questions on figure 5. Meanwhile, could the author provide high resolution figures for figure 3 B and C?

6.       The author forgot putting figure 6 in the manuscript.

7.     However, there is another publication, “Identification of the collagen type 1 alpha 1 gene (COL1A1) as a candidate survival-related factor associated with hepatocellular carcinoma” in  BMC, 2014.  So, could the author also give some discussion related to the study, since there are controversial conclusions between the two studies?

Author Response

Q1: Reviewer #2:   In the study, the author identified COL1A1 as a survival-related biomarker and therapeutic target in HCC and its metastasis.  Combining the deep data mining from TCGA, clinic samples check and cell in vitro study, the work showed high level of COL1A1 has a correlation to HCC and its metastasis. However, there are some important major and minor issues that should be fixed before this article could consider for publication in Cancers.

A1: We sincerely thank the reviewer for the time taken to review our work, and for the suggestions given. In this revised manuscript, we have made use of the reviewer’s critiques and suggestions, and hope this allay the reviewer’s concerns.

Q2: Reviewer #2:   The author should better add the dilution of primary antibody used in western blot study part.

.A2: We thank the reviewer for this comment. We have now indicated the dilution for all antibodies used. Please kindly see our revised Materials and Methods section, Pages 3-4, Lines 136-153.

2.5. Western blotting analysis

Cellular protein lysates were isolated using Protein Extraction Kit (QIAGEN, USA), and quantified by Bradford Protein Assay Kit (Beyotime, Beijing, China). An equal amount of (20μg) of total protein lysate sample was loaded per lane and subjected to sodium dodecyl sulfate polyacrylamide gel electrophoresis (SDS-PAGE). Separated proteins were transferred onto polyvinylidene fluoride (PVDF) membranes, followed by unwanted signal blocking using skim-milk in Tris-buffered saline (TBS), 1X PBS washing 3 times, and then incubation of the PVDF membranes with primary antibodies against COL1A1 (sc-293182, 1:1000), E-cadherin (sc-71008, 1:1000), Slug (sc-166476, 1:1000), Vimentin (sc-80975, 1:1000), purchased from santa cruz (Santa Cruz Biotechnology, Inc, Santa Cruz, CA, USA), and KLF4 (#4038S, 1:1000), OCT4 (#2890S, 1:1000), CD133 (#86781S, 1:1000), and β-actin (#4970S, 1:1000) from CST (Cell Signaling Technology, Inc, Danvers, MA, USA) at 4 oC overnight. The membranes were then incubated in goat anti-rabbit (1:10000; Jackson ImmunoResearch, West Grove, PA, USA) or anti-mouse (1:10000, BD Biosciences, San Jose, CA, USA) horseradish peroxidase (HRP)-conjugated secondary antibodies, and visualized using the enhanced chemiluminescence reagents (ECL, Amersham Biosciences, GE Healthcare, Chicago, IL, USA). The relative band intensity was analyzed using NIH ImageJ software (https://imagej.nih.gov/ij/) and expressed as the ratio of expressed protein to β-actin which served as loading control.

Q3: Reviewer #2:   Some writing might be modified, such as Protein-protein interaction (PPI) interaction analysis, people usually wrote as Protein-protein interaction (PPI) analysis.

A3: We appreciate this comment by the reviewer. We have now corrected this orthographical error. Please kindly see our revised Results section, Page 5, Lines 204-235.

3.1. COL1A1 is highly expressed in HCC and confers significant survival disadvantage

To investigate the COL1A1 expression profile in HCC and non-tumor liver tissue samples, we used the RNA sequencing and microarray gene profiling data (GSE14323/GPL571, GSE3500, GSE14520 and GSE6764) from GEO and TCGA. The TCGA data included 438 HCC samples, while the GSE14323 had 38 HCC, 58 cirrhosis and 19 normal liver tissue samples. Gene expression profile analysis using heatmap generated from the GSE14323 dataset showed 543 up-regulated genes in cirrhosis, and 89 down-regulated genes in HCC, with 87 of these genes commonly expressed in both cirrhosis and HCC (Figure 1A,B). Protein-protein interaction (PPI) analysis of top 21 common genes in cirrhosis and HCC was performed using the STRING functional protein association networks (https://string-db.org), showed that collagen type genes, including COL1A1 interact with molecular effectors of cytokine signaling in the immune system, namely chemokines CXCR4, CXCL6, CXCL9, CXCL10, CCL19, ANXA1, major histocompatibility complex (MHC) class II proteins HLA-DMA, HLA-DQA1, HLA-DPA1, HLA-DRA, HLA-DQB1, and peptide ligand-binding receptors PTPRC, IFIT1, IFI27 and IFI44L with an average local clustering coefficient of 0.724 and PPI enrichment p-value <1.0e-16 (Figure 1C). Gene expression analysis of paired tumor-non-tumor samples from TCGA-LIHC cohort (n = 438) also showed 1.13- (p<0.001), 1.01- (p<0.01), and 1.07- (p<0.001) fold reduction in COL1A1, COL1A2 and COL4A1 mRNA expression in the HCC tumor compared to the non-tumor ‘normal’ liver samples (Figure 1D). Furthermore, using median gene expression for threshold definition in our Kaplan-Meier survival analysis, we demonstrated that patients with high COL1A1 expression exhibited worse overall survival (OS) rate with a significant survival disadvantage (~18 - 26%, p = 0.049) compared to the low expression group, however, though statistically insignificant, the high COL1A2 and COL4A1 groups with ~12% (p = 0.410) and ~4% (p = 0.762) survival disadvantage, respectively, compared to the low expression groups (Figure 1E). To corroborate these findings, using the TMU-SHH HCC cohort (n = 72), we carried out a Cox proportional hazard analysis of COL1A1 expression and known risk factors for survival in patients with HCC, including age, hepatitis B surface antigen (HBsAg), cirrhosis, tumor size, TNM tumor stage, -fetoprotein (AFP), and lymph node involvement. Univariate analysis showed that age > 60, presence of cirrhosis, tumor size ≥20 mm, AFP ≥ 400ng/dL, lymph node metastasis, advanced TNM tumor stage, and high COL1A1 expression were risk factors for poor prognosis in the patients with HCC; and interestingly, multivariate analysis revealed that akin to age, cirrhosis, tumor size, AFP level, and TNM tumor stage, COL1A1 is an independent prognostic factor of OS in patients with HCC (Table 2).

Q4: Reviewer #2:   Meanwhile, about the PPI network analysis, could the author try 2 to 3 different interactome analysis tool to verify the results or confirm the top genes interaction?

A4: We thank the reviewer for this comment. We do agree with the reviewer on the use of more than one interactome analysis tools and confirm that we indeed did exactly that before choosing the most representative. Below are the ones we used in the course of this study:

(i)         String -  http://string-db.org/

(ii)      Protein Interaction Network Analysis (PINA):  Wu, J., Vallenius, T., Ovaska, K., Westermarck, J., Makela, T.P. and Hautaniemi, S. (2009) Integrated network analysis platform for protein-protein interactions, Nature methods, 6, 75-77.

(iii)    SAINT v2: Choi, H., Larsen, B., Lin., Z.-Y., Breitkreutz, A., Mellacheruvu, D., Fermin, D., Qin, Z.S., Tyers, M., Gingras, A.-C. and Nesvizhskii, A.I. (2011) SAINT: probabilistic scoring of affinity purification - mass spectrometry data. Nature Methods, 8:70-3.

(iv)     HIPPIE  -    http://cbdm.mdc-berlin.de/tools/hippie/

Q5: Reviewer #2:   2.2 Analyses of cancer microarray and RNAseq dataset, could the author add a little more detail on the data extraction and analysis?

A5: We are grateful for reviewer’s comment. As suggested, we have now addressed this in our revised manuscript. Please kindly see our revised Materials and Methods section, Page 3, Lines 101-120.

2.2. Analyses of cancer microarray and RNAseq dataset

COL1A1 gene expression profiling and correlative studies were performed using the Gene Expression Omnibus (GEO) human hepatocellular carcinoma microarray dataset with accession numbers GSE14323, GSE3500, GSE14520 and GSE6764 in the Oncomine platform (https://www.oncomine.org/resource/), and TCGA Liver Cancer hepatocellular carcinoma (LIHC) cohort (n = 373). All clinical data were downloaded from the TCGA portal using the University of California Santa Cruz (USCS) Xena functional genomics explorer (https://xenabrowser.net/heatmap/#) and survival analysis carried out.

Microarray and RNAseq dataset analyses were performed as previously described [21]. Briefly, after the pre-processing and microarray data statistical analyses using R statistical computing environment in the RStudio software version 1.0.143 (https://www.rstudio.com/), we processed the Affymetrix human gene 1.0 ST [HuGene-1_0-st] LIHC array datasets in computable document format (CDF)  (hugene10st_Hs_ENTREZG), to extract the most complete gene metadata annotation for the affymetrix probe identifier (IDs). This was followed by data normalization using the Robust Multi-array Average (RMA) algorithm, with log2-transformation and quartile-normalization of the datasets. Where multiple probes had same Ensembl gene identifier, median gene expression was used. Empirical Bayes-based linear models for microarray data using the limma r-package (http://bioconductor.org/packages/release/ bioc/html/limma.html) were employed for identification of differentially expressed genes (DEGs) and Benjamini-Hochberg procedure was used to adjust the p-values and reduce the false discovery rate (FDR).

 Q6: Reviewer #2:   About 2.6 colony formation assay, what’s the cell concentration plated in the 6 well plates? The author should better add a little more detail here.

A6: We thank the reviewer for this comment. We have taken the reviewer’s concern into consideration in our revised manuscript. Please kindly see our revised Materials and Methods section, Page 4, Lines 154-160.

2.6. Colony formation assay

To determine the colony-forming ability, 2.5 x 103 siCOL1A1-transfected or wild-type HCC cells were plated in triplicates in 6-well plates (Corning, Corning, NY, USA) consisting of a base layer of 0.5% agarose gel and an upper layer of 0.35% agarose gel with DMEM/F-12 medium, N2 supplement, 20 ng/mL of EGF, and bFGF and incubated for 7 days. The colonies formed were stained with 0.1% crystal violet in 20% methanol, counted and comparative analysis performed. In this study, we defined a colony as a cluster of ≥ 50 HCC cells.

Q7: Reviewer #2:   In line 279, MMP2 (0.40 (p = 8.29e-16)), should be MMP9 (0.40 (p = 8.29e-16)). Please check line 170, (bFGF; (Invitrogen, Grand Island, NY, USA),

.A7: We thank the reviewer for this observation. We have corrected this in our revised manuscript. Please kindly see our revised Results section, Page 9, Lines 302-316.

3.4. COL1A1 is strongly associated with and is a probable bridge between the metastatic and CSC-like phenotypes of HCC

Having established a critical role for COL1A1 in the induction and/or enhancement of HCC cells invasion and migration through EMT deregulation, in vitro, we investigated probable links between COL1A1-induced metastatic phenotype and CSC-like phenotype using mRNA data (RNA Seq V2) from the provisional TCGA LIHC cohort consisting of 371 patients and 373 samples. Results of our Spearman correlation analysis revealed strong correlation between COL1A1 and metastasis markers, vimentin (VIM) (0.79 (p = 1.25e-79)), snail (SNAI1) (0.75 (p = 6.21e-68)), matrix metalloproteinase (MMP)-2 (0.87 (p = 1.18e-116)), MMP9 (0.40 (p = 8.29e-16)), TWIST1 (0.67 (p = 6.53e-50)) (Figure 4A), as well as markers of cancer stemness, CD144/PROM1 (0.47 (p = 1.56e-21)), CD44 (0.39 (p = 3.03e-15)), KLF4 (0.37 (p = 1.46e13)), and ABCC1 (0.50 (p = 1.39e-24)) (Figure 4B). However, we found no significant correlation between COL1A1 and SOX2 (0.07 (p = 0.193)) (Figure 4B). These bioinformatics-based statistical data do indicate a strong association between COL1A1, markers of metastasis and cancer stemness, as well as suggest a role for COL1A1 in the modulation of HCC CSCs-like phenotype.

Q8: Reviewer #2:   Please add detail on IHC staining, including the primary and secondary antibody dilution, incubation time, et al. meanwhile, could the author put clinic sample data detail as supplementary table in the manuscript, including patient age, sex, risk factors (virus infection, or alcohol or NASH etc)?

A8: We thank the reviewer for this observation. We have included the requested detail in our revised manuscript. Please kindly see our revised Materials and Methods section, Page 4, Lines 175-187.

2.9. Immunohistochemistry

Standard immunohistochemistry (IHC) protocol was used for gene expression profiling and analysis non-tumor liver and HCC tissue samples harvested from the Taipei Medical University-Shuang Ho Hospital HCC cohort (TMU-JIRB No. 201302016). Briefly, 5μm-thick sections were first de-waxed using xylene for 5 min twice and re-hydrated with 100% ethanol twice for 5 min, 95% ethanol for 5 min and 80% ethanol for 5 min, followed by blocking of endogenous peroxidase activity using 3% hydrogen peroxide. Antigen retrieval process was carried out in a pressure cooker where the slides were immersed in 10 mmol/L ethylenediaminetetraacetic acid (EDTA) (pH 8.0) for 3 min, followed by blocking with 10% normal serum. The sections were then incubated with anti-COL1A1 antibody (1:200) overnight at 4 OC, followed by goat anti-mouse IgG HRP-conjugated secondary antibody (1:10000) from the HRP Polymer Kit (#TP-015-HD; Lab Vision, Fremont, CA, USA). Diaminobenzidine (DAB) was used as the chromogenic substrate, and sections were counterstained with Gill's hematoxylin (Fisher Scientific, NJ, USA).

Q9: Reviewer #2: For “clinic sample data details” as requested by the reviewer. Please kindly see our Tables section, Tables 1 and 2.

Table 1. Patient clinicopathological characteristics of TMU-SHH HCC cohort

Table 2. Univariate and multivariate analysis of COL1A1 expression in TMU-SHH HCC cohort

Please also kindly see our revised Materials and Methods section, Pages 2-3, Lines 92-100.

2.1. Ethics approval and consent to participate

This study was conducted in a cohort of patients with HCC at Taipei Medical University Shuang-Ho Hospital, Taipei, Taiwan. The study was reviewed and approved by the institute review board (TMU-IRB-2018-0001). In our representative TMU-SHH HCC samples (n = 72), patients aged from 36 to 81 with median age of 68.0; 45 were male (62.5%) while 27 were female (37.5%). Based on clinical data extracted from archived patients' demographic, clinical and biochemical investigation information, clinical characteristics including HBsAg status, cirrhosis, TNM stage, tumor size and lymph node involvement are also presented in Table 1. The median follow-up time was 26.1 months and 7 patients died during follow-up.

Also kindly see our revised Results section, Pages 5-6, Lines 204-235.

3.1. COL1A1 is highly expressed in HCC and confers significant survival disadvantage

To investigate the COL1A1 expression profile in HCC and non-tumor liver tissue samples, we used the RNA sequencing and microarray gene profiling data (GSE14323/GPL571, GSE3500, GSE14520 and GSE6764) from GEO and TCGA. The TCGA data included 438 HCC samples, while the GSE14323 had 38 HCC, 58 cirrhosis and 19 normal liver tissue samples. Gene expression profile analysis using heatmap generated from the GSE14323 dataset showed 543 up-regulated genes in cirrhosis, and 89 down-regulated genes in HCC, with 87 of these genes commonly expressed in both cirrhosis and HCC (Figure 1A,B). Protein-protein interaction (PPI) analysis of top 21 common genes in cirrhosis and HCC was performed using the STRING functional protein association networks (https://string-db.org), showed that collagen type genes, including COL1A1 interact with molecular effectors of cytokine signaling in the immune system, namely chemokines CXCR4, CXCL6, CXCL9, CXCL10, CCL19, ANXA1, major histocompatibility complex (MHC) class II proteins HLA-DMA, HLA-DQA1, HLA-DPA1, HLA-DRA, HLA-DQB1, and peptide ligand-binding receptors PTPRC, IFIT1, IFI27 and IFI44L with an average local clustering coefficient of 0.724 and PPI enrichment p-value <1.0e-16 (Figure 1C). Gene expression analysis of paired tumor-non-tumor samples from TCGA-LIHC cohort (n = 438) also showed 1.13- (p<0.001), 1.01- (p<0.01), and 1.07- (p<0.001) fold reduction in COL1A1, COL1A2 and COL4A1 mRNA expression in the HCC tumor compared to the non-tumor ‘normal’ liver samples (Figure 1D). Furthermore, using median gene expression for threshold definition in our Kaplan-Meier survival analysis, we demonstrated that patients with high COL1A1 expression exhibited worse overall survival (OS) rate with a significant survival disadvantage (~18 - 26%, p = 0.049) compared to the low expression group, however, though statistically insignificant, the high COL1A2 and COL4A1 groups with ~12% (p = 0.410) and ~4% (p = 0.762) survival disadvantage, respectively, compared to the low expression groups (Figure 1E). To corroborate these findings, using the TMU-SHH HCC cohort (n = 72), we carried out a Cox proportional hazard analysis of COL1A1 expression and known risk factors for survival in patients with HCC, including age, hepatitis B surface antigen (HBsAg), cirrhosis, tumor size, TNM tumor stage, α-fetoprotein (AFP), and lymph node involvement. Univariate analysis showed that age > 60, presence of cirrhosis, tumor size ≥20 mm, AFP ≥ 400ng/dL, lymph node metastasis, advanced TNM tumor stage, and high COL1A1 expression were risk factors for poor prognosis in the patients with HCC; and interestingly, multivariate analysis revealed that akin to age, cirrhosis, tumor size, AFP level, and TNM tumor stage, COL1A1 is an independent prognostic factor of OS in patients with HCC (Table 2).

Q9: Reviewer #2:   About TCGA data, did the author combine all various HCC samples, if choose only one, there usually showed about 373- 377 samples in total?

A9: We thank the reviewer for this comment. While we are not sure we understand what the reviewer’s question is, we understand the necessity of data integrity and reproducibility, and thus we confirm the veracity of our all data used in our study. Having provided details of means and methods used for our GEO and TCGA datasets, the reviewer may freely verify.

Q10: Reviewer #2:   Please add high resolution picture of the figure 1c. and the figure legend 1 E is not clear, the author should better modify it. About the figure 1B, could the author put the full gene list as supplementary table in the manuscript?

A10: We appreciate the reviewer’s suggestions. As requested, we have provided a high resolution picture for Figure 1C. Please kindly see our attached high resolution file for Figure 1C.

Q11: Reviewer #2:   To figure 2C, could the author double check the figures to make sure the figures used correctly, especially on the HCC late stage.

A11: We sincerely thank the reviewer for this observation. For HCC late stage, image place as x200 was actually for x400 and vice versa. We have now re-arranged the data appropriately in our revised manuscript. Please kindly see our updated Figure 2C and its legend, Page 8, Lines 269-275.

Figure 2. Upregulated COL1A1 expression at both mRNA and protein levels strongly correlates with disease progression. (A) Differential expression of COL1A1 in HCC (upper panel) and cirrhosis (lower panel) compared to normal liver samples in the GSE14323 Mas Liver dataset. (B) COL1A1 is highly expressed in GSE3500 Chen Liver , GSE14520 Roessler Liver, and GSE6764 Wurmbach Liver HCC specimens compared to non-tumor liver specimens. (C) Representative IHC photo-image (left) and graphical quantification (right) of COL1A1 protein expression in early stage, late stage HCC tissues, and adjacent normal tissues. *p < 0.05, **p < 0.01, ***p < 0.001.

Q12: Reviewer #2:   To the figure 3 study, what’s the control, the author should better marked clearly, is it the cell treated with scramble siRNA or the cell without any treatment?  There are similar questions on figure 5. Meanwhile, could the author provide high resolution figures for figure 3 B and C?

A12: We thank the reviewer for this suggestion. We have clarified the term “control (Ctl)” used in our revised manuscript. Please kindly see our revised Results section, Page 9, Lines 295-301.

Figure 3. Knockdown of COL1A1 suppressed HCC cell migration and invasion through deregulated EMT, in vitro. (A) Western blot images showing the knockdown efficacy of COL1A1 in SNU-387 or SNU-475 cells. Representative photo-images (left) and histograms (right) of the effect of siCOL1A1 on the (B) invasion and (C) migration of SNU-387 and SNU-475 cells. (D) Representative western blot images of the effect of siCOL1A1 on the expression levels of COL1A1, E-cadherin, Slug and vimentin protein in SNU-387 and SNU-475. β-actin was used as a loading control. *p < 0.05, **p < 0.01, ***p < 0.001; Ctl, wild type cells not transfected with siCOL1A1

Also kindly see our revised Results section, Page 11, Lines 349-357.

Figure 5. Loss of COL1A1 function significantly impair colony and tumorsphere formation of HCC cells in vitro. Representative photo-image (left) and histograms (right) showing the inhibitory effect of siCOL1A1 on the (A) colony formation potential and (B) number and size of primary and secondary hepatospheres formed by SNU-387 or SNU-475 cells. (C) Western blot images of the effect of siCOL1A1 on the expression level of COL1A1, KLF4, OCT4, and CD133 proteins. (D) Functional characterization and druggability studies using the SAMSUNG HCC, ACBG HCC, and TCGA LIHC cohorts. Arrowhead, gene of interest; Dr, Small molecule drugability score; TS, Tumor suppressor; On, Oncogene; Im, involved in Immune modulation; Se, Secreted gene; Su - Cell Surface protein. β-actin was used as a loading control. *p < 0.05, **p < 0.01, ***p < 0.001; Ctl, wild type cells not transfected with siCOL1A1

Q13: Reviewer #2:   The author forgot putting figure 6 in the manuscript.

A13: We sincerely thank the reviewer for pointing out this omission. We have now included the pictorial abstract (Figure 6) in our revised manuscript. Please kindly see our included Figure 6 and its legend, Page 13, Lines 429-431.

Figure 6. Pictorial abstract showing that the therapeutic or molecular targeting of COL1A1 function significantly impair the oncogenicity and cancer stem cell-like phenotype of HCC cells by deregulating EMT and abrogating HCC stemness.

Q14: Reviewer #2:   However, there is another publication, “Identification of the collagen type 1 alpha 1 gene (COL1A1) as a candidate survival-related factor associated with hepatocellular carcinoma” in BMC, 2014.  So, could the author also give some discussion related to the study, since there are controversial conclusions between the two studies?

A14: We thank the reviewer for this insightful comment. We have now addressed the issue raised by the reviewer in our revised manuscript. Please kindly see our revised Discussion section, Page 11-12, Lines 358-427.

4. Discussion

The development of synchronous or metachronous metastasis by patients with HCC even post curative surgical resection is a medical challenge in liver oncology clinics. While majority of these cases are intrahepatic metastatic, it is estimated that about 13.5% - 42% are extrahepatic metastasis for patients with HCC [25 - 28]. The relatively limited therapeutic options for managing metastatic and late stage HCC [25 - 28] necessitates the discovery of reliable targetable or druggable disease-specific biomarkers such as COL1A1, and gives clinical and/or translational relevance to the findings of this present study. The present study provides preclinical evidence indicating that COL1A1 is a reliable biomarker and putative therapeutic target for hepatocellular carcinogenesis and metastasis by demonstrating that (i) COL1A1 is highly expressed in HCC and confers significant survival disadvantage, and that (ii) upregulated COL1A1 expression at both mRNA and protein levels strongly correlates with disease progression. We also showed that the (iii) knockdown of COL1A1 suppressed HCC cell migration and invasion through deregulated EMT, in vitro. Finally, we demonstrated that (iv) COL1A1 is strongly associated with and is a probable bridge between the metastatic and CSC-like phenotypes of HCC, and that the (v) loss of COL1A1 function significantly impair colony and tumorsphere formation of HCC cells in vitro.

Our findings that COL1A1 is highly expressed at both mRNA and protein levels in HCC, strongly correlates with disease progression, and confers significant survival disadvantage to patients with HCC (Figures 1 and 2) is consistent with recent findings showing that COL1A1, activated by multiple signaling pathways in a myocardin-related transcription factor A (MRTFA)-mediated manner, is highly expressed in human breast cancer [29], as well as suggestion that COL1A1 and COL1A2 mRNA levels are significantly upregulated in gastric cancer tissues compared to normal tissues, and that lower COL1A1 and COL1A2 expression levels was strongly associated with better overall survival in patients with gastric cancer [30]. However, this demonstrated implication of high COL1A1 expression in poor prognosis among patients with HCC contradicts findings from an isolated study suggesting that COL1A1 gene expression is significantly decreased in HCC samples and that this tumor-associated suppressed COL1A1 mRNA expression levels significantly correlated with poor OS [31]. While we cannot fully rationalize this contrasting conclusion, it is worth pointing out that in same study, COL1A1 methylation and expression analysis of the “study patient” and HCC cell lines showed apparent enhancement of COL1A1 methylation status, mRNA and protein expression levels; this is antithetic to the study conclusions and relevant in the context of contemporary knowledge wherein DNA methylation not only predisposes tissue cells to tumorigenesis, but is increasingly shown to contribute to the tumor state via repression of tumor suppressor genes and inhibition of cellular differentiation plasticity [32, 33]; by same token, there is evidence that persistent methylated DNA after tumor resection indicates residual disease and is associated with disease recurrence and poor prognosis [reviewed in 33].

Furthermore, concordant with the accruing evidence that enhanced expression of COL1A1 promotes metastasis of breast cancer [34], augment the proliferation and invasion of gastric cancer cells by downregulating miR-129-5p [35], and via dysregulation of the WNT/PCP pathway, promotes metastasis in colorectal cancer (CRC), our finding that the knockdown of COL1A1 suppressed HCC cell migration and invasion by downregulating slug and vimentin, while upregulating E-cadherin, in vitro (Figure 3) is not out of place. In fact additional corroboration of our finding is found in the work of Toiyama Y, et al [36] showing that while slug and vimentin are highly expressed in higher tumor (T) stage, lymph-node involvement, hepatic metastasis and advanced TMN stage, siRNA-silencing of Slug in CRC cells induced reduction in cell proliferation, suppressed EMT, and attenuated the invasion and migration in CRC cells.

In line with recent findings that enhanced invasiveness and clonogenicity characterize CSCs, and that circulating liver CSCs play critical roles in the preparation of new sites for malignant colonization [23, 24], having demonstrated that COL1A1 plays a critical role in the induction and/or enhancement of HCC cells invasion and migration through the deregulation of EMT, in vitro, we investigated probable links between COL1A1-induced metastatic phenotype and CSCs-like phenotype. Our results revealed that COL1A1 is strongly associated with and is a probable bridge between the metastatic and CSC-like phenotypes of HCC (Figure 4), which is particularly interesting and bear translational relevance against the background that metastatic disease is implicated in over 90% of cancer related mortalities, and in the light of the role of CSCs in the self-renewal of cancerous cells, resistance to anti-cancer therapy, metastatic dissemination to secondary sites, and disease recurrence [37]. Based on our findings, we posit that aberrant expression of COL1A1 confer enhanced clonal survivability on enriched cancerous cells and drive the acquisition of cellular traits that favor the formation and dissemination of micro- and macro-metastases culminating the intra-tumoral Darwinian evolutionary bioprocess. This position is reinforced by the fact that the loss of COL1A1 function significantly impair colony and tumorsphere formation of HCC cells in vitro (Figure 5), and consistent with current therapeutic strategies that target CSCs-like HCC cells with enhanced capacity to colonize distant organs and exert tumor-initiating potential in the tumor microenvironment [22]. Interestingly, using bioinformatics-based algorithms for functional characterization and druggability studies, we also showed that COL1A1 is a targetable or druggable oncogene that not only has a high degree of gene neighborliness with, but is a probable modulator of other collagen types such as COL1A2, COL4A1, and COL4A5, survival genes ABL1, ABL2, and PPARG, as well as known markers of cancer stemness, namely ALDH1, CD44, CD133/PROM1, KLF4, MYC, OCT4/POU5F1, and SOX2, in patients with HCC (Figure 5).
